# Imaging Carotid Plaque Inflammation Using Positron Emission Tomography: Emerging Role in Clinical Stroke Care, Research Applications, and Future Directions

**DOI:** 10.3390/cells12162073

**Published:** 2023-08-15

**Authors:** John J. McCabe, Nicholas R. Evans, Sarah Gorey, Shiv Bhakta, James H. F. Rudd, Peter J. Kelly

**Affiliations:** 1Health Research Board Stroke Clinical Trials Network Ireland, Catherine McAuley Centre, Nelson Street, D07 KX5K Dublin, Ireland; sarahgorey@mater.ie (S.G.); pjkelly@mater.ie (P.J.K.); 2Neurovascular Unit for Applied Translational and Therapeutics Research, Catherine McAuley Centre, Nelson Street, D07 KX5K Dublin, Ireland; 3School of Medicine, University College Dublin, Belfield, D04 V1W8 Dublin, Ireland; 4Stroke Service, Department of Medicine for the Elderly, Mater Misericordiae University Hospital, Eccles Street, D07 R2WY Dublin, Ireland; 5Department of Clinical Neurosciences, Box 83, Addenbrooke’s Hospital, University of Cambridge, Hills Road, Cambridge CB2 0QQ, UK; ne214@cam.ac.uk (N.R.E.); sab201@cam.ac.uk (S.B.); 6Division of Cardiovascular Medicine, Addenbrooke’s Hospital, University of Cambridge, Hills Road, Cambridge CB2 0QQ, UK; jhfr2@cam.ac.uk

**Keywords:** inflammation, stroke, prognosis, positron emission tomography, recurrence, carotid

## Abstract

Atherosclerosis is a chronic systemic inflammatory condition of the vasculature and a leading cause of stroke. Luminal stenosis severity is an important factor in determining vascular risk. Conventional imaging modalities, such as angiography or duplex ultrasonography, are used to quantify stenosis severity and inform clinical care but provide limited information on plaque biology. Inflammatory processes are central to atherosclerotic plaque progression and destabilization. 18F-fluorodeoxyglucose (FDG) positron emission tomography (PET) is a validated technique for quantifying plaque inflammation. In this review, we discuss the evolution of FDG-PET as an imaging modality to quantify plaque vulnerability, challenges in standardization of image acquisition and analysis, its potential application to routine clinical care after stroke, and the possible role it will play in future drug discovery.

## 1. Introduction

Stroke is a leading global cause of death and disability. Every year, 12 million people have a stroke and 6.5 million die due to stroke. One in four adults will have a stroke during their lifetime [1]. Despite modern guideline-based prevention therapy, the risk of major adverse cardiovascular events (MACE) after stroke approaches 30% at 5 years [2]. There are over 100 million stroke survivors worldwide and the global economic cost of stroke is USD 891 billion (1.1% of global GDP) [1]. The burden of stroke in Europe is forecast to increase by one-third over the next 30 years [3]. These figures underscore the global health importance of stroke prevention.

In conventional clinical practice, risk stratification for recurrence after stroke in the setting of carotid atherosclerosis is guided by the severity of luminal stenosis [4]. The evidence informing this approach is based on findings from randomized control trials (RCTs) published over thirty years ago, which compared best medical therapy (BMT) and carotid endarterectomy (CEA) with BMT alone [5]. BMT was variably defined in these trials as antiplatelet treatment and, as indicated, antihypertensive, lipid-lowering, and glucose-lowering medications. In pooled analyses of these RCTs in symptomatic carotid disease, patients with severe stenosis (70–99% luminal narrowing) had the greatest reduction (absolute risk reduction (ARR) 16%) in stroke risk with CEA, with more modest benefit (ARR 4.6%) observed in patients with moderate stenosis (50–69% luminal narrowing) [5]. Patients with mild stenosis (<50%) did not benefit from intervention. Guidelines therefore recommend that patients with moderate stenosis are selected for revascularization on the basis of a favourable risk–benefit assessment, but do not recommend CEA in patients with <50% stenosis [4]. However, there are several unresolved questions with respect to patient selection for revascularization. First, the RCTs were performed over thirty years ago and BMT has evolved substantially during that time. Second, there was significant heterogeneity in the benefit accrued from CEA in patients with moderate stenosis according to sex, age, time to intervention, and index mechanism [6]. Third, irrespective of luminal stenosis severity, plaque vulnerability can now be better characterized using advanced imaging modalities to identify high-risk patients [7,8,9,10]. Therefore, improved methods to identify patients at the highest risk of stroke are needed to refine selection for carotid revascularization, allowing surgery to be targeted towards patients most likely to benefit.

Incorporating an assessment of plaque vulnerability may provide important prognostic information regarding the recurrence risk in patients with symptomatic carotid atherosclerosis irrespective of the degree of luminal stenosis. Plaque rupture is the final common pathway leading to the majority of thrombo-embolic events in patients with carotid atherosclerosis. In the Oxford Plaque Study of 526 patients who underwent CEA for symptomatic carotid stenosis (≥50%), the histological features associated with fibrous cap rupture included intraplaque haemorrhage (IPH), large lipid-rich necrotic core (LRNC), thin fibrous cap, and marked macrophage cap infiltration [11]. In recent years, imaging techniques to improve the identification of high-risk plaque have illustrated the importance of capturing information on the biological vulnerability of carotid plaque for risk stratification purposes. The presence of micro-embolic signals (MES) is a marker of asymptomatic cerebral emboli and is associated with a marked increase in stroke risk in patients with symptomatic carotid disease [7]. Plaque neovascularization imaged by contrast-enhanced carotid ultrasound is associated with several other markers of plaque vulnerability including plaque rupture [8], and has also been associated with recurrent stroke [9]. High-resolution carotid plaque MRI (HR-MRI) can identify several features of high-risk plaque including IPH, LRNC, and thin fibrous plaques, all of which are strongly associated with recurrent stroke [10].

However, inflammation is also central to the initiation, progression, and destabilisation of atherosclerotic plaque [12]. Clinico-pathological data clearly demonstrate the association between carotid plaque macrophage content, plaque vulnerability, and recurrence risk [13,14]. Positron emission tomography (PET) imaging of the carotid artery can also identify features of plaque vulnerability [15]. 18F-fluorodeoxyglucose (FDG) is a radio-labelled glucose analogue which is avidly taken up by metabolically active macrophages and accumulates at tissue sites of increased inflammatory activity. FDG-PET has now become a validated imaging technique for quantifying plaque inflammation. Improving imaging techniques to evaluate the presence and severity of carotid plaque inflammation after stroke is attractive for several reasons. First, new imaging biomarkers might identify patients with vulnerable atherosclerotic plaque who may be at risk of recurrent stroke. Second, it might provide additional prognostic information beyond the degree of luminal stenosis and guide the selection of patients for carotid revascularization. Third, it may also be useful to identify patients who may benefit from targeted anti-inflammatory therapy after stroke.

In this review, we will (1) provide an overview of the contribution of atherosclerosis to the global burden of stroke; (2) discuss the biology of immune mechanisms in atherosclerosis; (3) review the technical aspects of image acquisition/analysis and the limitations of PET as an imaging modality; (4) explore the utility of FDG-PET for determining recurrence risk in symptomatic carotid disease; (5) outline its potential use as a surrogate endpoint in future RCTs of anti-inflammatory therapy after stroke; and (6) discuss the potential future evolution in PET imaging in atherosclerosis using novel radio-labelled tracers.

## 2. The Importance of Atherosclerosis in Stroke

Atherosclerotic mechanisms are central to the pathogenesis of many stroke subtypes. Large artery atherosclerotic disease is a clearly defined mechanism in approximately 13–17% of stroke cases [16]. Extracranial carotid artery stenosis is the commonest cause of large artery atherosclerotic strokes in Caucasians [17]. Large artery atherosclerosis of the intracranial vasculature is responsible for approximately 30–56% of ischaemic stroke/transient ischemic attack (TIA) in Asia [18], and although less frequent in Blacks, Hispanics, and Caucasians, intracranial atherosclerotic disease (ICAD) remains an important cause of stroke in these ethnicities [17,19]. Patients with large artery atherosclerosis have also been shown to carry a higher risk of early recurrent stroke than other stroke subtypes [16,17] and the mere presence of atherosclerosis in any vascular bed confers a two- to threefold increased risk of late-outcome major vascular recurrence [20].

Cryptogenic stroke represents about 25% of all ischaemic stroke cases, but the role of atherosclerotic mechanisms in such patients is increasingly apparent [21]. The presence of any non-stenotic carotid plaque is reportedly found in 79% of patients with embolic stroke of uncertain source (ESUS) [22]. High-risk plaque features are also more frequently identified ipsilateral to a cryptogenic infarct compared with the contralateral carotid [23,24,25]. Atherosclerotic disease of the aortic arch is more prevalent in patients with cryptogenic stroke than patients with known stroke mechanisms and is also associated with a higher risk of stroke recurrence [26,27]. Finally, high-resolution 7-Tesla MRI can identify a high proportion of culprit intracranial plaque in patients otherwise meeting the diagnostic criteria for cryptogenic stroke [28]. It is also likely that atherosclerotic mechanisms also play a role in the pathogenesis of lacunar infarcts in the deep white matter. Fisher described autopsy evidence of “microatheroma” or foam cell accumulation in the walls of the parent or feeding vessels of these patients, causing stenosis or occlusion [29,30]. Irrespective of the precise role of atherosclerosis in small vessel occlusion, patients with lacunar stroke frequently have a high burden of coronary atherosclerosis on autopsy [31] and coronary events represent approximately one-quarter of all major vascular events after lacunar stroke [32]. Atherosclerosis is also a frequent cause of stroke in patients who experience stroke despite anticoagulation for atrial fibrillation [33] and is associated with a high risk of recurrence in this patient group [34].

## 3. The Biology of Atherosclerosis

Over the past twenty years, our understanding of the complex mechanisms underlying plaque biology have deepened considerably. Inflammation is important in atherosclerosis [35]. An immune process is involved at every stage of plaque development. Leukocyte accumulation in the arterial intima follows endothelial expression of adhesion molecules, such as vascular cell adhesion molecule-1 (VCAM-1) [36]. This is triggered by a variety of atherogenic stimuli including dyslipidaemia and hypertension [37]. Oxidised low-density lipoprotein (LDL) in the arterial wall is a potent pro-inflammatory stimulus which upregulates the expression of adhesion molecules and chemokines. Leukocyte migration into the intima is dependent on the expression of chemokines such as monocyte chemoattractant protein-1 (MCP-1) and interleukin-8 (IL-8) [38,39]. Recruited monocytes differentiate into macrophages, which phagocytose LDL, forming lipid-laden foam cells [40]. Resident plaque macrophages are instrumental to the propagation of the inflammatory process in atherosclerosis. Monocytes and macrophages release reactive oxygen species and cytokines, including interleukin-1β (IL-1β) and tumour necrosis factor-α (TNFα), which promote the inflammatory cascade within the plaque. T-cell lymphocytes are also involved early in plaque development by stimulating phagocytes to express tissue factor, matrix metalloproteinases (MMPs), and pro-inflammatory cytokines [12]. Later, smooth muscle cells migrate into the intima and proliferate, a process which is facilitated by platelet-derived growth factor (PDGF). Smooth muscle cells ultimately form collagen and fibrin, which culminates in fibrous plaque formation [12]. As the plaque develops, apoptotic macrophages leave behind an extracellular accumulation of lipids and cellular debris. This ineffective clearance of dead cells and their contents forms what is commonly referred to in the literature as the ‘lipid-rich necrotic core’ (Figure 1) [40].

Plaque progression is an uneven process. The fibrous cap overlying atheromatous plaque can erode through overexpression of MMPs which degrade collagen. Pro-inflammatory cytokines released from resident macrophages/foam cells promote MMP expression in smooth muscle, endothelial cells, and mononuclear phagocytes. Thin fibrous caps are vulnerable to rupture, which exposes the underlying thrombogenic lipid-rich plaque core to clotting factors, triggering thrombus formation [35]. This can be followed by the clinical sequelae of myocardial infarction or stroke. However, more commonly, mural thrombi heal through the propagation of anti-inflammatory and fibrinolytic mechanisms, a process that may lead to the formation of a calcified fibrous plaque. Advanced plaques often present with microvessel formation, which develop in response to angiogenic factors, such as vascular endothelial growth factor (VEGF), which is released by macrophages [41]. This neovasculature is friable and prone to bleeding, which can result in intraplaque haemorrhage; one of the hallmarks of unstable plaques. Intraplaque haemorrhage stimulates smooth muscle migration and plaque expansion [12]. Hence, plaque development is a dynamic process, where the balance between pro-inflammatory and repair mechanisms dictate plaque stability.

Robust data link inflammation to prognosis in stroke. Histo-pathological data from clinical studies of patients with resected carotid plaque also clearly demonstrate the association between carotid plaque macrophage content and plaque vulnerability [13,14]. The Oxford Plaque Study reported a robust association between the severity of plaque inflammation, plaque macrophage content, and plaque instability, ulceration, and thrombosis [13]. Data from the Dublin Carotid Atherosclerotic Stroke Study (DUCASS) also showed that extensive macrophage and lymphocytic infiltration in resected carotid plaques was independently associated with early recurrent stroke [14]. Blood inflammatory markers, such as high-sensitivity c-reactive protein (hsCRP) and interleukin-6 (IL-6) are independently associated with first stroke [42,43] and recurrent vascular events after ischemic stroke/TIA [44].

## 4. Development of FDG-PET Imaging as a Marker of Plaque Vulnerability

The observation that atherosclerotic plaques show increased FDG pooling on whole-body PET scans led researchers to test the hypothesis that PET imaging could identify unstable atherosclerotic lesions. In 2002, Rudd et al. reported that FDG uptake was 27% higher in symptomatic carotid plaques when compared with the contralateral side in eight patients who subsequently underwent CEA [15]. Animal models subsequently illustrated the association between FDG uptake and macrophage density in excised plaque. Using an animal model of atherosclerosis (Watanabe heritable hyperlipidemic (WHHL) rabbits), Ogawa et al. showed that FDG accumulates in atherosclerotic plaque [45]. In their experiment, 13 WHHL and 3 control rabbits were injected with FDG, and underwent PET-CT imaging, followed by thoracic/abdominal aortas excision 4 h later. FDG uptake was significantly greater in the aortas of the WHHL rabbits than in those of the control rabbits and very strongly correlated (R^2^ = 0.83) with plaque macrophage number on histo-pathological specimens [45]. Similar work undertaken on 17 patients with symptomatic carotid stenosis undergoing CEA showed that FDG uptake in vivo strongly correlated with plaque macrophage area (R^2^ = 0.68) and the % CD68 staining (a macrophage specific monoclonal-antibody) (R^2^ = 0.70) of the co-registered resected plaque sections [46]. The FDG signal imaged by PET also correlated with the gene expression of markers of plaque vulnerability and increased metabolic activity, including CD68, cathepsin K, IL-18, GLUT-1, and hexokinase (HK2) [47,48,49]. A significant weakness of this early work was the limited accuracy of in vivo PET to co-localize the FDG signal with histo-pathological findings, calling into question the assumption that FDG uptake reflected macrophage hypermetabolism as opposed to the metabolic activity of other cellular components of plaque. However, in a study utilizing high-resolution microPET on resected human plaques, thirteen patients undergoing CEA were injected with FDG one hour before the procedure, and ex vivo MRI-PET imaging was performed 4 h after plaque resection. This study convincingly demonstrated that increased FDG uptake was associated with plaque inflammatory cell infiltration (R^2^ = 0.32, *p* = 0.001) and neovascularization (R^2^ = 0.24 *p* = 0.02), but was inversely associated with calcification (R^2^ = −0.32, *p* < 0.001) [50]. It is possible that FDG uptake is not only a marker of plaque inflammation, but is also increased in the presence of hypoxia, which is a feature of complex atherosclerotic plaque. Pre-clinical work demonstrated that the glucose metabolism of monocyte macrophages is increased in the setting of hypoxia, but not in the presence of pro-inflammatory cytokines [51,52]. However, one study demonstrated that hypoxia-inducible factor-1α (HIF-1) gene expression is only very weakly correlated (R^2^ = 0.19) with the FDG signal and the association was no longer significant when CD68 was entered into the model as a predictor of FDG uptake [47].

## 5. PET Imaging Acquisition and Analysis

Vascular PET imaging involves several technical considerations, including tracer-specific factors, quantification methods, and approaches to co-registration. These considerations are important not only for optimizing the detection of ‘atheroinflammation’, but also for ensuring reproducible results in the case of longitudinal imaging for measuring responses to pharmacological intervention.

The nature of FDG as an analogue of glucose has important implications for PET imaging acquisition, with circulating glucose competing with FDG for cellular uptake via facilitative transport. Reflecting this, pre-scan glucose levels are positively associated with higher blood pool FDG activity, and negatively associated with uptake in carotid atheroma, both of which may bias the measurement of tracer activity in the region of interest [53]. In order to minimize this effect from circulating glucose, participants are required to fast prior to FDG-PET. The EANM has made recommendations on this issue, stating that glucose levels should ideally be <7 mmol/L prior to PET imaging. Recommended approaches to managing instances with sub-optimal glucose levels include administration of rapid-acting insulin prior to injection of FDG or asking the patient to hydrate while ambulating and recheck the blood glucose level periodically until an acceptable level has been achieved [54]. If glucose levels remain >7.0 mmol/L, a dedicated formula is suggested for normalizing the measured glucose content for an overall population average of 5.0 mmol/L [54]:SUVgluc = SUV × patient’s blood glucose in milligrams per decilitre (mmol/L)/90 mg/dL
(5.0 mmol/L)

Although highly sensitive (where picomolar tracer concentrations of the tracer can be detected), the non-specific uptake of FDG may cause technical challenges. The resolution of PET (typically around 3 mm) can result in partial volume effects, where uptake in neighbouring structures has the potential to spill over into regions of interest. Strategies, such as reducing vocalization during tracer uptake periods to reduce physiological uptake in the vocal cords, may mitigate such effects. Similarly, it may be very challenging to measure tracer uptake in regions of interest next to highly metabolically active structures, such as the difficulty in evaluating the coronary arteries due to FDG uptake in the adjacent myocardium that spills over to obscure the uptake within coronary atherosclerosis. Consequently, there is increased attention to developing PET ligands with greater specificity for targeting physiological processes of interest, that results in superior signal-to-noise ratios (see below).

Differences in acquisition protocols and analysis have meant that comparisons between studies and performing meta-analyses can be challenging. Consequently, there have been calls for standardized methodologies. The European Association of Nuclear Medicine (EANM) Cardiovascular Committee aimed to address this by producing a position paper recommending optimized and standardized protocols for the imaging and interpretation of PET scans in atherosclerosis [54]. These recommendations include a circulation time of two hours and an injected activity of FDG of 3–4 MBq/kg. Furthermore, the position paper recommends arterial imaging should ideally be performed in participants with pre-scan glucose levels lower than approximately 130 mg/dL (approximately 7–7.2 mmol/L).

Techniques to measure tracer uptake in arterial disease can be divided into two broad approaches: the standardized uptake value (SUV) and tissue-to-background ratio (TBR). SUV—the standard approach to measuring tracer uptake within solid organs—is calculated as the ratio of radiotracer concentration within the target tissue to the injected radiotracer activity adjusted for weight. The TBR takes this one step further by correcting for blood pooling of the radiotracer, as calculated by the ratio of the SUV of the arterial wall to the SUV in the mid-luminal region within the venous system [55].
SUV_max_ = maximum activity in the region of interest (mBq/g)/(injected dose [mBq]/body weight [g])
TBR_max_ = SUV_max_/jugular vein blood pool SUV_mean_

Both approaches have strengths and weaknesses. In a methodological study considering the SUV and TBR approaches to measuring tracer uptake in individuals with carotid atherosclerosis, only TBR readings (and not SUV readings) were significantly different between inflamed versus non-inflamed plaques on histological analysis of endarterectomy specimens [56]. However, TBR may also be affected by lower renal clearance (resulting in a higher blood pool SUV) and longer injection-to-scan intervals (associated with decreasing blood pool SUV) [57,58]. The EANM Cardiovascular Committee’s position paper recommends quantification of FDG uptake using TBR instead of SUV, given that the use of a ratio between two measurements limits the effects on signal quantification of errors in patient weight, the dose of radiotracer injected, and of the imaging time point [54]. However, as outlined in Section 7, the association with recurrence after stroke is stronger and more robust for SUV rather than TBR. The precise reasons for this are unclear. Based on the available evidence, the authors’ view is that SUV is the preferred parameter for quantifying stroke recurrence risk in patients with carotid atherosclerosis.

Quantification of tracer uptake may be further considered according to the maximum tracer uptake within a region of interest (resulting in SUV_max_ or TBR_max_) or mean tracer uptake within the region of interest (SUV_mean_ and TBR_mean_) [55]. Furthermore, different studies may measure the tracer uptake along the whole vessel, within the most diseased segment (as a measure of plaque uptake), or in active segments (where only arterial regions with tracer uptake above a pre-specified value are considered) (Figure 2). For example, in Elkhawad et al., a TBR threshold of ≥1.6 was used to define an active segment, as uptake below this value was associated with atheroma with <5% inflammation within the plaque [47,59]. In the analysis of culprit carotid plaque metabolic activity after acute stroke, the single hottest slice (SHS) is often taken as the measure of interest for defining plaque inflammation and is defined as the axial slice with the highest SUV_max_ within the plaque segment.

Accurate measurement of tracer uptake in the region of interest requires PET to be co-registered with an appropriate structural imaging modality. Historically, this has been achieved through hybrid imaging of PET with computed tomography (PET/CT), but more recent developments in hybrid PET/magnetic resonance imaging (PET/MRI) means that this approach is increasingly used in research. PET/MRI has advantages over PET/CT in terms of improved soft tissue contrast due to high spatial resolution. This is particularly relevant for being able to image high-risk plaque morphology (thin fibrous cap, intraplaque hemorrhage, large lipid-rich/necrotic cores) alongside measurement of PET tracer uptake. However, this technique also carries novel challenges: hybrid PET/MRI systems required adaptation to ensure detection equipment was compatible with the magnetic field, whilst also avoiding electromagnetic disruption to the magnetic field. Furthermore, existing attenuation correction developed for PET/CT utilized attenuation maps generated from tissue density measured directly by CT. MRI is unable to measure tissue density directly, and consequently attenuation correction has relied on tissues being separated into four classes (air, lungs, fat, water), where this approach has been found to produce PET information of similar quality to attenuation correction using low-dose CT scans [60,61]. Finally, any benefits of the hybrid PET/MRI systems need to balanced alongside several practical considerations, including the limited availability of the hybrid systems, increased cost, non-compatibility with indwelling metal implants, and tolerability.

## 6. Vascular Inflammation Imaged by PET and Systemic Vascular Risk

Early clinical work [15] demonstrating that the FDG signal is a marker of recently symptomatic carotid plaques was replicated in several studies. A systematic review showed that FDG uptake, quantified as either SUV_max_ or TBR_max_, was higher in recently symptomatic carotid atherosclerotic plaques compared with the asymptomatic carotids [62]. TBR_max_ in one artery is strongly associated with TBR_max_ in neighbouring arteries, suggesting that plaque inflammation measured by PET might be a surrogate marker of a systemic inflammatory atherosclerotic plaque burden [63].

Two large studies have reported that the association between background plaque inflammation imaged by FDG-PET in patients without a history of vascular disease is independently associated with incident major vascular events, including stroke [64,65]. In a Korean cohort, 1089 middle-aged adults invited for health screening underwent PET-CTA imaging [64]. The common/internal carotid artery FDG signal was quantified as TBR_max_ at each slice, with an average “whole-vessel” reading over serial slices taken as a measure of background vascular inflammation. FDG uptake was quantified as “high” or “low” according to an arbitrary threshold of >75th vs. ≤75th percentile. Patients with high TBR_max_ had a higher incidence of cardiovascular events (3.3% vs. 1.0%) during a median follow-up of 4.2 years. Elevated arterial inflammation, quantified as whole-vessel TBR_max_, was associated with MACE after adjustment (hazard ratio (HR) 2.98, 95% CI 1.17–7.62, *p* = 0.01). When incorporated into the Framingham risk score (FRS), TBR_max_ improved the identification of incident cardiovascular events compared with FRS alone (area under the curve (AUC) 73.2 vs. 59.9, *p* = 0.04) with a net reclassification improvement (NRI) of 40.1% [64]. These data suggested that background arterial inflammation might be an independent risk factor for future vascular events in apparently healthy adults and might improve risk stratification beyond that of the FRS. However, the study had several important limitations: the potential for selection bias, a retrospective design, a high male predominance (95%) of included patients, the small number of outcome events suggesting the analyses were under-powered, uncertainty regarding incomplete adjustment for potentially confounding cardiovascular risk factors, and the definition of the endpoint, which included silent myocardial ischemia and TIA. In another study, of 613 cancer-free patients without a vascular history undergoing PET-CT imaging for oncological surveillance, TBR_max_ measured in the ascending aorta was again independently associated with vascular events after adjustment for age, smoking, hyperlipidemia, and hypertension (HR 4.71; 95% CI 1.98 to 11.2, *p* < 0.001, top vs. bottom tertile TBR_max_) [65]. Similar findings were found when SUV_max_ was taken as the exposure variable (HR 2.67, 95% CI 1.02 to 6.97; *p* = 0.04), top vs. bottom tertile). The addition of TBR to the FRS was also shown to improve the identification of incident vascular events (c-statistic 0.66 for FRS + TBR vs. 0.62 with FRS alone), with a significant improvement in the NRI for the 10% intermediate-risk threshold (27.5%, 95% CI: 16.5 to 39.9). The study had a high proportion of patients with a history of prior malignancy (83%), which limited the generalizability of its findings [65]. Nevertheless, both studies showed consistent results, demonstrating that systemic vascular inflammation is associated with increased vascular risk in patients without a history of cardiovascular disease in mid-life. Moreover, the observed risk was independent of other cardiovascular risk factors and the magnitude of the effect was relatively large.

The ongoing Progression of Early Subclinical Atherosclerosis (PESA) study has prospectively enrolled over 700 patients to an advanced imaging substudy using hybrid PET-MRI. Eligible patients have evidence of atherosclerotic plaque identified on ultrasound screening during mid-life. This study will provide important insights into the role of plaque and arterial inflammation in atherosclerosis progression and destabilization [66]. Early findings from this group have shown that background arterial inflammation is common (48%) in people with subclinical atherosclerosis, but often does not co-locate to sites of atherosclerotic plaque. In fact, just 11% of plaques identified on MRI showed evidence of a hypermetabolic state. The risk factors most consistently associated with arterial inflammation included age, male sex, smoking, and obesity [67].

## 7. Carotid Plaque Inflammation and Stroke Risk

In patients with carotid atherosclerosis, several cross-sectional imaging studies have demonstrated that FDG-PET provides additional information regarding plaque vulnerability, beyond the degree of luminal stenosis. These studies show that plaque inflammation is associated with other imaging biomarkers of plaque instability identified using MRI, CTA, carotid contrast-enhanced ultrasonography, and transcranial Doppler. FDG uptake was substantially higher in carotid lesions with evidence of high-risk morphological features characterized by multidetector CTA. TBR_mean_ was greater in patients with evidence of plaque remodelling (median [IQR]: (3.0 [2.17–3.71] vs. 1.40 [1.27–2.0], *p* = 0.001) and low-attenuation plaque (3.0 [1.85–3.71] vs. 1.50 [1.26, 2.0], *p* = 0.001) [68]. Plaque inflammation is also associated with features of complicated plaque (defined as American Heart Association (AHA) type VI) in patients with cryptogenic stroke imaged using carotid HR-MRI. Type VI plaque demonstrated a higher FDG signal (TBR = 3.43 ± 1.13) in comparison with other AHA lesions (TBR 2.41 ± 0.84). FDG uptake was also higher in ruptured atherosclerotic plaques and in plaques with a thin fibrous cap, as compared with non-ruptured lesions with thick fibrous caps (TBR = 3.55 ± 1.21 and 3.14 ± 1.05 vs. 2.38 ± 0.83; *p* < 0.001) visualized on MRI. Similarly, plaques with LRNC were more inflamed than those without this finding (TBR 3.14 ± 1.14 vs. 2.36 ± 0.80, *p* < 0.001) and patients with IPH on MRI also showed significantly higher FDG uptake as compared with other lesions (3.48 ± 1.11 vs. 2.40 ± 0.84, *p* < 0.001) [69]. Similar findings were reported by Giannotti et al. in a cohort of patients with recently symptomatic carotid artery stenosis who underwent combined PET and MRI imaging [70]. SUV_max_ strongly correlated with plaque LRNC in the corresponding axial slice (r_s_ = 0.64, *p* < 0.001) and was inversely associated with whole-plaque fibrous cap thickness (r_s_ = −0.4, *p* = 0.02) and calcium volume (r_s_ = −0.4, *p* = 0.03) [70]. TBR_max_ was also found to be higher in patients with MES detected on transcranial Doppler than those without MES. These findings extended to patients with symptomatic and asymptomatic carotid stenosis [71,72].

Until recently, there was little data on the association between plaque hypermetabolism identified on FDG-PET CTA and recurrent stroke in patients with symptomatic internal carotid stenosis. In 2012, our research group reported an association between whole-vessel carotid SUV_max_ and the risk of early ipsilateral recurrence after adjustment for luminal stenosis in the DUCASS study [73]. In a subsequent individual-participant-data meta-analysis of 196 patients from three prospective cohort studies (DUCASS, Biomarkers/Imaging Vulnerable Atherosclerosis in Symptomatic Carotid Disease (BIOVASC), and the Barcelona Plaque Study), SUV_max_ in the single hottest slice of the symptomatic ICA was independently associated with early recurrent stroke after adjustment for cardiovascular risk factors and stenosis severity [74]. Per g/mL increase in SUV_max_ there was an approximate twofold increased risk of early recurrent stroke (HR 2.19, 95% CI, 1.41–3.39; *p* < 0.001) [74]. These findings have since been replicated in another study performed in Singapore with similar eligibility criteria [75]. More recent work demonstrated that FDG uptake in the SHS is also associated with 5-year ipsilateral stroke recurrence even after accounting for carotid revascularization and stenosis severity (HR 1.98, 95% CI 1.10–3.56, *p* = 0.02, per g/mL) [76].

## 8. Carotid PET Imaging for Risk Stratification after Stroke

Prognostic information from measuring stenosis severity to guide selection of patients for carotid revascularization is well validated [5]. However, combining information on plaque inflammation and luminal stenosis into a single score has been shown to improve the identification of both early and late outcome recurrent ipsilateral stroke in patients with symptomatic carotid stenosis 50–99%. The symptomatic carotid atheroma inflammation lumen-stenosis (SCAIL) score (range 0–5) includes 3 points for FDG activity (SUV_max_ < 2 g/mL, 0 points; SUV_max_ 2–2.99 g/mL, 1 point; SUV_max_ 3–3.99 g/mL, 2 points; SUV_max_ ≥ 4 g/mL, 3 points) and 2 points for stenosis severity (<50%, 0 points; 50–69%, 1 point; ≥70%, 2 points) (Table 1). In an individual participant data meta-analysis of three prospective studies, SCAIL was independently associated with early recurrent stroke (HR 2.40, 95% CI 1.2–4.5, *p* = 0.01, per 1-point increase). Compared with stenosis severity alone (c-statistic 0.63, 95% CI, 0.46–0.80), the prediction of recurrent stroke was improved with the SCAIL score (c-statistic 0.82, 95% CI 0.66–0.97, *p* = 0.04 for comparison) [77]. More recently, the same findings were reported for the outcome of 5-year recurrent ipsilateral stroke (adjusted HR 2.73, 95% CI 1.52–4.90, *p* = 0.001 per 1-point increase in SCAIL) [76]. When the score was collapsed into low- (≤2) and high-risk (3–5) SCAIL categories, the 5-year ipsilateral recurrent stroke risk was 2.6% in patients with a score of <3 vs. 14.3% in patients with a score of ≥3. A SCAIL score of ≥3 had 88.2% sensitivity and 45.1% specificity for discrimination of 5-year recurrent stroke [76]. These data clearly demonstrate that plaque inflammation imaged by FDG-PET is strongly associated with both early- and late-outcome recurrent stroke in patients with symptomatic carotid stenosis. Figure 3 and Figure 4 provide two case examples of the application of PET-CT imaging in stroke care.

These studies also show that measuring plaque inflammation provides additional prognostic information for recurrence beyond that of luminal stenosis and might be useful in guiding the selection of patients for revascularization. In a pooled analysis of published trials of CEA vs. medical therapy in patients with symptomatic carotid atherosclerosis, several subgroups of patients did not demonstrate benefits from revascularization [6]. No benefit was observed in patients with mild stenosis (<50%), in those who presented with ocular symptoms (regardless of stenosis severity), and in important subgroups with moderate stenosis (women, age < 65 years, patients with TIA symptoms at onset, diabetes, and delayed CEA > 2 weeks) [6]. Consequently, the guidelines recommend the selection of patients with moderate stenosis for revascularization based on a careful risk–benefit assessment [4]. In a study exploring the predictive utility of FDG-PET in patients with uncertain benefit from revascularization, the SCAIL score was again independently associated with early recurrent stroke [78]. The analysis was restricted to patients with luminal stenosis < 50%, retinal TIAs, or moderate stenosis with “low-risk” criteria as outlined above. A SCAIL score ≥ 3 had a sensitivity of 72.2% and a specificity of 62.4% for discrimination of stroke recurrence, and a score of ≥2 had a sensitivity of 100% [78]. Again, these data support the concept that FDG-PET can help improve patient selection for CEA or stenting. Randomized trials that select patients for revascularization based on PET imaging findings might further the case for its routine use in clinical practice. Limitations to the clinical application of PET imaging in this scenario include the limited availability of PET in many centres and the cost implications. However, it can also be argued that a more selective approach to CEA might reduce the costs associated with unnecessary surgery, hospitalization, and recurrent stroke.

## 9. PET Imaging in Randomized Control Trials

Randomized control trials of anti-inflammatory therapies in coronary artery disease provide compelling evidence that targeting inflammatory mechanisms can reduce vascular risk. A meta-analysis of four RCTs showed that colchicine reduces the risk of MACE and halves the risk of stroke in patients with CAD [79]. Inhibition of the IL-6 signaling pathway with the interleukin-1β (IL-1β) antagonist canakinumab also reduces the risk of MACE in coronary disease [80]. Colchicine is now under evaluation for the secondary prevention of vascular events after stroke in the CONVINCE trial (NCT02898610). Despite several positive trial results, RCTs of anti-inflammatory therapies for prevention in cardiovascular disease are at an early stage, particularly after stroke. A pipeline of new tailor-made anti-inflammatory agents, developed specifically for vascular prevention, are likely to come onstream in the coming years. However, definitive phase 3 RCTs demonstrating clinical efficacy require large sample sizes, are expensive, and take several years to design and complete. To efficiently evaluate new agents, surrogate endpoints in RCTs to demonstrate potential efficacy are needed for anti-inflammatory therapies. Vascular PET imaging has potential translational utility for clinical trial design. Changes in plaque hypermetabolism identified on PET imaging could be used as a surrogate endpoint in phase 2 RCTs, and sample size calculations indicate that a relatively small numbers of participants with a short follow-up would be sufficient to detect significant effects [81]. For example, the pleiotropic anti-inflammatory effects of statins were previously illustrated in a clinical trial using the PET signal as an endpoint. In this double-blind RCT, low-dose atorvastatin was compared with high-dose atorvastatin in patients at high cardiovascular risk. Participants receiving 80 mg atorvastatin had significantly greater reductions in the arterial FDG signal compared to baseline (−14%) when compared with the 10 mg dose (−4%). Changes in plaque inflammation were seen as early as 4 weeks after randomization and were not correlated with the degree of LDL reduction [82].

The strengths of using the PET signal as a surrogate endpoint are (i) plaque inflammation identified on PET imaging is strongly associated with the clinically relevant endpoint that one would use in definitive trials (i.e., recurrent stroke); (ii) there is overwhelming evidence that inflammation lies on the causal pathway between plaque instability and recurrent events [37,76]; (iii) plaque hypermetabolism is very stable when measured over time and inter-/intra-observer agreement is excellent [81]; (iv) the FDG signal is an objective biological endpoint and not subject to physician- or patient-dependent reporting biases; (v) standardized protocols exist for image acquisition and analysis [54]; (vi) candidate therapies that do not demonstrate efficacy in adequately powered phase 2 RCTs using PET may not warrant further evaluation in phase 3 trials and this should reduce research waste. However, it should be acknowledged that uncertainties remain regarding the use of PET for this purpose [83]. First, for the simple reason that the question has never been studied, it is unknown if the anti-inflammatory effects of FDG lowering will reduce clinical outcome events, or indeed what magnitude of FDG reduction would be clinically meaningful. However, we postulate that large reductions (>10%) in the FDG signal would be required to convincingly demonstrate whether the drug in question would have a reasonable likelihood of reducing recurrent stroke risk. Future studies validating the use of the FDG-PET signal as a surrogate endpoint in RCTs are needed and would greatly strengthen the rationale for its application in the realm of emerging anti-inflammatory therapies. Second, the ‘off-target’ effects of anti-inflammatory agents would not be captured by PET imaging endpoints alone and, therefore, monitoring for safety endpoints (e.g., neutropenia, thrombocytopenia, infection) in such RCTs would be mandated.

## 10. Novel Tracers and Future Directions

Although FDG has proven an effective radiotracer for imaging atherosclerosis, the ubiquitous uptake of FDG within any metabolically active tissue means that measurement of its uptake in atheroma may be affected by spill-over from adjacent structures. Hence, the development of new radiotracers with specificities that match the high sensitivity of the imaging technique has the potential to improve atherosclerosis imaging through superior signal-to-noise ratios. Furthermore, emerging radiotracers targeting different pathophysiological processes within the atherosclerotic plaque may also aid our understanding of atherogenesis in vivo, as well as facilitate identification of the vulnerable plaque (Figure 1) (Table 2).

Gallium-68-labelled [1,4,7,10-tetraazacyclododecane-*N*,*N*′,*N*″*N*‴-tetraacetic acid]-D-Phe^1^ Tyr^3^-octreotate (DOTATATE) is a ligand with a high specificity for the somatostatin receptor subtype-2 (SST_2_) that is upregulated on the surface of activated macrophages [84,85]. DOTATATE has been used successfully to image inflammation within atherosclerotic plaques, with excellent macrophage specificity (r = 0.89, 95% confidence interval 0.28–0.99, for *SSTR2* gene expression that is seen exclusively in pro-inflammatory M1 macrophages) and superior performance compared to FDG in terms of imaging coronary atherosclerosis (where DOTATATE uptake was seen only in the atherosclerotic plaque and recently infarcted myocardium, but not diffusely within the myocardium as seen with FDG) [86,87]. In individuals undergoing ^64^Cu-DOTATATE-PET imaging for neuroendocrine tumours, ^64^Cu-DOTATATE uptake in the carotid arteries was significantly increased in the presence of cardiovascular risk factors compared to those without risk factors, and individuals with known cardiovascular disease had significantly higher SUV_max_ than those with vascular risk factors but no cardiovascular disease (mean difference 0.30 g/L, *p* = 0.02) [88].

Ligands for the CXC motif chemokine receptor 4 (CXCR4) have also shown promise for evaluating plaque inflammation. CXCR4 is involved in the trafficking of progenitor and inflammatory cells to the vulnerable plaque, within which CXCR4-positive cells are seen to aggregate [89]. CXCR4 has been seen to co-localize with CD68 expression on immunofluorescence [90]. ^68^Ga-pentixafor is a specific CXCR4 ligand, and its arterial uptake, quantified as TBR, increases with the increasing burden of cardiovascular risk factors and calcification [91,92]. The TBR_max_ of pentixafor uptake was also increased in vulnerable carotid plaques, demonstrating eccentricity compared to non-eccentric plaque morphology [90].

Activated macrophages highly express translocator protein (18 kDa) (TSPO) [93]. In vitro studies have found that PK11195 uptake co-localized with macrophage-rich regions in excised carotid atherosclerotic plaques [94,95], and that uptake of PK11195 was higher in symptomatic carotid plaques than in asymptomatic plaques (TBR 1.06 ± 0.20 versus 0.86 ± 0.11, respectively) [96].

Other macrophage-related markers have also been investigated for identifying inflamed—and consequently vulnerable—atherosclerotic plaques. αvβ3 integrin is a transmembrane glycoprotein expressed by CD68-positive macrophages and endothelial cells [97]. αvβ3 expression in carotid atherosclerosis has been targeted using ^18^F-Galacto-RGD, where TBR correlated with αvβ3 expression in excised tissue, and was significantly higher in stenotic versus non-stenotic areas of the carotid artery [98]. ^68^Ga-NODAGA-RGD is an alternative tracer for αvβ3 integrin, and its use has shown that uptake in the arterial wall of major arteries was significantly increased compared to those without (mean TBR 2.44 [2.03–2.55] versus 1.81 [1.56–1.96], *p* = 0.001), and correlated with plaque burden (rho = 0.31, *p* = 0.04) and prior cardiovascular or cerebrovascular events (r = 0.33, *p* = 0.027) [99].

Metabolic processes involved in atherogenesis may also be exploited for evaluating the degree of inflammation in an atherosclerotic plaque. Cellular proliferation, such as that seen for macrophages within the inflamed atherosclerotic plaque, involves the uptake of choline, a precursor of phosphatidylcholine that is required for the synthesis of cellular membranes [100]. In a study of men undergoing ^11^C-choline imaging for prostate cancer, FDG uptake, calculated as SUV_max,_ was common in the carotids and aorta, with most of the tracer uptake occurring in areas of non-calcified atherosclerosis [101]. In a proof-of-principle study of 10 individuals undergoing carotid endarterectomy, ^18^F-fluorocholine (FCH) uptake, quantified as TBR_max_, was significantly higher in symptomatic atheroma versus contralateral carotid arteries, and correlated with plaque CD68 content (rho = 0.648, *p* = 0.043), but did not correlate with the degree of carotid artery stenosis [102].

^11^C-acetate PET has been used to image fatty acid synthesis in the atherosclerotic vessel wall: uptake was seen frequently in individuals undergoing oncological imaging, where 88.8% had at least one site of uptake. Similar to FCH, uptake of ^11^C-acetate in areas of macrocalcification was seen only rarely [103].

Inflammation within the plaque may be associated with other important pathophysiological processes contributing to plaque rupture. Microcalcification—the formation of small deposits of calcium less than 50 μm in diameter within the fibrous cap [104]—occurs secondary to inflammation-driven differentiation of vascular smooth muscle cells to an osteoblast-like phenotype. These microcalcific deposits may in turn propagate and exacerbate inflammation [105]. ^18^F-sodium fluoride (NaF) is a radioligand that has been used for bone imaging since 1962, and has subsequently found clinical use in the PET evaluation of osseous metastatic disease [106]. NaF identifies sites of active microcalcification, where radio-labelled fluoride is exchanged for the hydroxyl group in hydroxyapatite to form fluoroapatite [107]. Vascular uptake of NaF was first reported by Derlin et al., who found incidental arterial uptake in 57 (76%) of 75 individuals undergoing whole-body NaF-PET for assessment for bone metastases. NaF uptake was seen in areas with and without macrocalcification, indicating that NaF uptake reflects an active microcalcification process rather than merely the burden of macrocalcification [108]. NaF arterial TBR is also correlated with the Framingham risk score [109]. Uptake of NaF in the coronary arteries (TBR) is increased in those with coronary artery disease compared to controls, in plaques with high-risk morphological features, and in more severe disease (as measured by symptom burden, troponin levels, or need for revascularization) [110,111]. On the other hand, the coronary arterial FDG signal was difficult to quantify due to myocardial spillover and was not associated with coronary atherosclerosis or culprit plaques [110,111]. TBR and SUV NaF uptake has been found to be higher in symptomatic versus asymptomatic carotid atherosclerotic plaques [112,113,114]. However, direct comparisons of NaF and FDG are limited. One small study reported that NaF, but not FDG, was associated with culprit carotid plaque status and plaque vulnerability [114]. However, one other study reported that whilst both FDG and NaF measured in the most-diseased arterial segment were both higher on the side of the symptomatic carotid plaque, only FDG was higher on a whole-vessel approach [113]. Therefore, NaF uptake is typically focally increased at the bifurcation compared to the more diffuse pattern of uptake seen with FDG [113]. Whether carotid NaF uptake is associated with an increased risk of recurrent neurovascular events remains unknown and represents an important avenue of future study.

## 11. Conclusions

Atherosclerosis is a major contributor to the global burden of stroke and an important determinant of vascular recurrence risk. Quantifying plaque vulnerability using FDG-PET has been shown to improve risk estimation in symptomatic carotid stenosis. This has the potential to inform patient management and selection for carotid revascularization. Heterogeneous approaches to PET image acquisition and analysis of vascular inflammation remain a barrier to standardization, but recent guidelines by the EANM are an important first step to alleviating this issue. FDG-PET imaging may have an important role as a surrogate endpoint in future RCTs of anti-inflammatory therapies as we move into a new era of drug development for prevention in cardiovascular disease. Novel tracers of plaque inflammation may have benefits over FDG but will need to be shown to improve the prediction of clinical endpoints after stroke.

## Figures and Tables

**Figure 1 cells-12-02073-f001:**
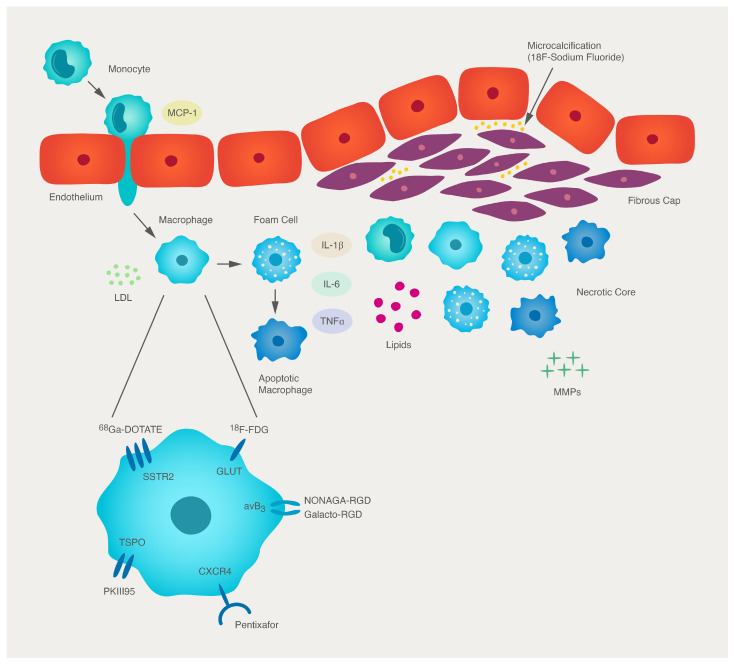
Schematic illustration of atherosclerotic plaque progression and PET radiotracer targets. Endothelial activation leads to the expression of leukocyte adhesion molecules. Chemokines, particularly MCP-1, are important in directing the intimal migration of monocytes. Resident monocytes acquire characteristics of a tissue macrophage, scavenging lipoprotein particles such as LDL. These processes give rise to the arterial foam cell, which is the result of the accumulation of lipid droplets within the cytoplasm. The foam cell secretes pro-inflammatory cytokines (IL-1β, IL-6, TNFα) that amplify the local inflammatory response. Microcalcification deposits form within the fibrous cap secondary to inflammation-driven differentiation of vascular smooth muscle cells to an osteoblast-like phenotype. MMPs can degrade extracellular matrix required for the integrity of the fibrous cap. When the plaque ruptures, blood comes into contact with tissue factor triggering thrombosis. Apoptotic macrophages contribute to the ‘necrotic core’ of the atherosclerotic lesion. Radiotracers can be used to quantify specific inflammatory processes within the plaque and be used as biomarkers of plaque vulnerability. FDG is a radio-labelled glucose analogue which is avidly taken up by metabolically active macrophages via GLUT transporters. ^68^Ga-DOTATE binds to SST2 receptors which are upregulated on the surface of activated macrophages. ^11^C-PK11195 is specific to the TSPO receptor. Pentixafor is a specific CXCR4 ligand involved in inflammatory cell trafficking to plaque. NaF is a radiotracer specific to microcalcification. αvβ3 integrin is expressed by CD68 macrophages, and radiotracers targeting this include NODAGA-RGD.

**Figure 2 cells-12-02073-f002:**
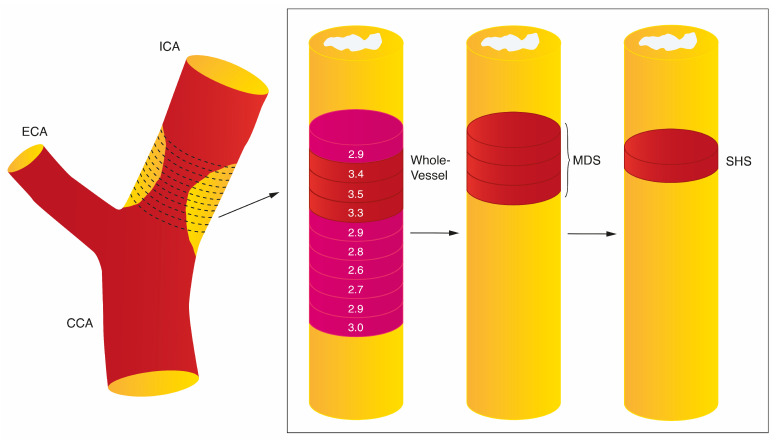
Schematic illustration of FDG-PET image analysis. Ten regions of interest (ROIs) are defined as 1 mm thick slices above and below the area of maximal luminal narrowing in the symptomatic carotid artery. FDG uptake in the vessel wall is quantified using standardized uptake values (SUV (g/mL)), with the highest uptake in each slice defined as the SUV_max_. The SHS is defined as the axial slice with highest SUV_max_. FDG uptake in the most diseased segment (MDS) is taken as the average of the SUV_max_ across three slices defined relative to the SHS. The whole-vessel approach takes an average of all arterial slices in the target vessel of interest. CCA, common carotid artery; ECA, extracranial carotid artery; ICA, internal carotid artery.

**Figure 3 cells-12-02073-f003:**
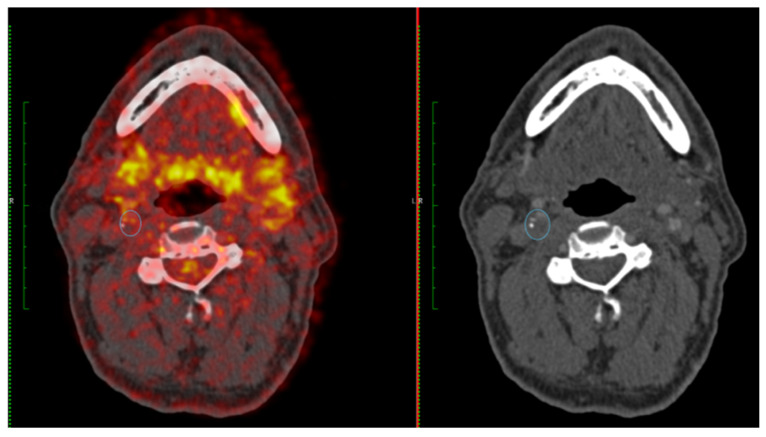
A 65-year-old female with right-sided amaurosis fugax. Duplex ultrasonography demonstrated a severe right internal carotid artery (ICA) stenosis (70–99%). FDG-PET (**left panel**) shows moderate FDG uptake in the right ICA (SUV_max_ 2.45 g/L) and CT angiography confirms a severe stenosis (**right panel**) (SCAIL score 3).

**Figure 4 cells-12-02073-f004:**
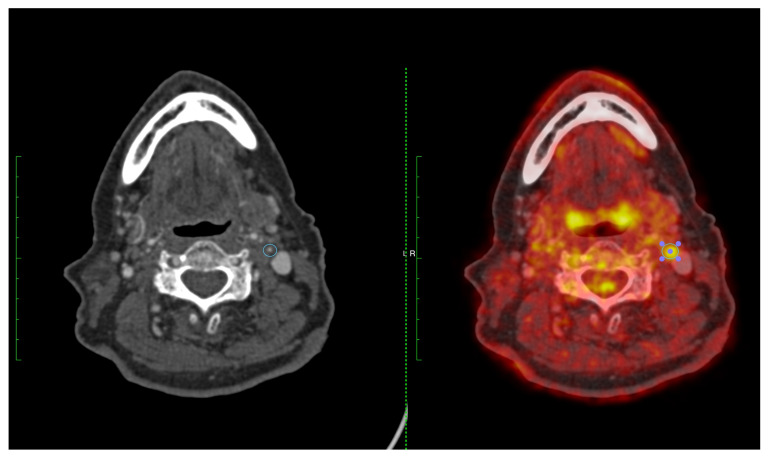
A 74-year-old man with expressive dysphasia and a left frontal infarct. CT angiography demonstrated a severe left ICA stenosis measuring 70–80% (**left panel**). FDG-PET showed high FDG uptake in the left ICA 4.05 g/L (**right panel**) (SCAIL score 5).

**Table 1 cells-12-02073-t001:** SCAIL score measures and points.

	Measure	SCAIL Points
Plaque SUV_max_, g/mL	<2	0
	2–2.99	1
	3–3.99	2
	≥4	3
Lumen stenosis, %	<50	0
	50–69	1
	≥70	2
Total		0–5

**Table 2 cells-12-02073-t002:** Novel PET radiotracers for imaging in atherosclerosis.

Ligand	Target
**Cellular targets**	
DOTATATE	Somatostatin receptor subtype-2 (SST_2_), which is upregulated on the surface of activate macrophages.
NODAGA-RGD and Galacto-RGD	αvβ3 integrin, expressed by CD68-positive macrophages and endothelial cells.
Pentixafor	Specific CXCR4 ligand, which is involved in the trafficking of inflammatory cells to the plaque.
PK11195	Translocator protein (18 kDa), expressed on activated macrophages.
**Metabolic targets**	
Acetate	Fatty acid synthesis.
Fluorocholine	Cellular proliferation (involving the uptake of choline).
Sodium fluoride	Microcalcification.

## Data Availability

No new data was created for this manuscript.

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
