# Peer review of "Imaging Carotid Plaque Inflammation Using Positron Emission Tomography: Emerging Role in Clinical Stroke Care, Research Applications, and Future Directions"

_cells, 2023, doi:10.3390/cells12162073_

Round 1

Reviewer 1 Report

The submitted review article is a valuable contribution to the field. Please respond to the below-mentioned comments/recommendations:

1. In line 25, in abstract section, 18F-fluorodeoxyglucose should be 18F-fluorodeoxyglucose.

2. Please provide citation for Line 42-43, 43-45.

3. In Line 46, please mention how the surgical benefit was quantified or measured?

4. In Line 47 and 48, please clarify the difference between risk and absolute risk? Please also clarify if % defines degree of stenosis.

5. In Line 49, what do you mean by "best medical therapy"?

6. Please substantiate the reason why improved methods are needed to identify patients as mentioned in Line 52-54. Provide citations to show that present methods for identifying high-risk patients are missing patients.

7. In Line 107, what do you mean by "TIA"

8. In Line 106-109, please clarify why large artery atheroscelerosis being less frequent is still considered important cause of stroke/

9. Provide citation to Line 178, Line 196-198,

10. Please clarify why in Line 198, strong correlation has R2 = 0.83 but in Line 205, R2=0.68 is still referred to showing strong correlation.

11. In Line 216, what is R2 for strong correlation between FDG uptake and macrophage/lymphocytic cellular infiltration?

12. On Line 220, what do you mean by "Metabolic activity". Please clarify.

13. In Line 223, why R2 = 0,19 is presented as a modest correlation? What is the range for poor, modest and strong correlation in terms of R2 values?

14. In Line 230, do you mean "tracer-specific activity" instead of "tracer-specific factors"?

15 Please add equation of both SUV and TBR to mathematically show how TBR calculation avoids consideration of patient's weight, injected dose and imaging time point.

16. What does author think if SUV or TBR should be used?

17. In entire review please clarify in publications cited the high PET signal or high uptake when comparing was quantified as SUV or TBR? For example, in Line 330-332, is FDG uptake quantified as SUV or TBR?

18. Provide citation to Line 339-341 and line 358-362.

19. In Line 405, provide citation for Gionnotti et al.

20. In SCAIL Points in Table 1 why TBR is not used if TBR is preferred over SUV?

21. Please provide TBR or SUV values when mentioning uptake of different tracers in section 10 for novel tracers, For example nu uptake values are given for DOTATATE in Line 559-567.

22. Please discuss studies where FDG is directly compared with the other novel tracers discussed in section 10.

23. Throughout the paper if all relevant citations are mentioned and there is no statement where an appropriate citation is not mentioned.

Author Response

Dear Reviewer,

Thank you for your comments. We have made significant changes to the manuscript after careful consideration of your comments.

1. In line 25, in abstract section, 18F-fluorodeoxyglucose should be 18F-fluorodeoxyglucose.

This has been amended

2. Please provide citation for Line 42-43, 43-45.

This has been amended

3. In Line 46, please mention how the surgical benefit was quantified or measured?

This has been amended

4. In Line 47 and 48, please clarify the difference between risk and absolute risk? Please also clarify if % defines degree of stenosis.

This has been amended

5. In Line 49, what do you mean by "best medical therapy"?

This has been amended

6. Please substantiate the reason why improved methods are needed to identify patients as mentioned in Line 52-54. Provide citations to show that present methods for identifying high-risk patients are missing patients.

This paragraph has been substantially amended

7. In Line 107, what do you mean by "TIA"

This has been amended

8. In Line 106-109, please clarify why large artery atheroscelerosis being less frequent is still considered important cause of stroke/

I thank the reviewer for their comment. However, I have not changed this paragraph. Large artery atherosclerotic is not substantially “less frequent” than other stroke subtypes. Stroke is a heterogenous disease with several other mechanisms also playing important roles. I think that the paragraph demonstrates that large artery atherosclerosis is an important cause of stroke: “Extracranial carotid artery stenosis is the commonest cause of large artery atherosclerotic strokes in Caucasians.16 Large artery atherosclerosis of the intracranial vasculature is responsible for approximately 30-56% of ischaemic stroke/TIA in Asia17, and although less frequent in Blacks, Hispanics and Caucasians, intracranial atherosclerotic disease (ICAD) remains an important cause of stroke in these ethnicities.16,18 Patients with large artery atherosclerosis have also been shown to carry a higher risk of early recurrent stroke than other stroke subtypes15,16 and the mere presence of atherosclerosis in any vascular bed confers a 2- to 3-fold increased risk of late-outcome major vascular recurrence.19

9. Provide citation to Line 178, Line 196-198,

This has been amended

10. Please clarify why in Line 198, strong correlation has R2 = 0.83 but in Line 205, R2=0.68 is still referred to showing strong correlation.

The BMJ (https://www.bmj.com/about-bmj/resources-readers/publications/statistics-square-one/11-correlation-and-regression) have provided arbitrary cut-offs for correlation coefficients which we will adhere to in the manuscript as follows:

<0.20 is very weak

0.20-0.39 is weak

0.40-0.59 moderate

0.60-0.79 is strong

0.80-1.00 is very strong

11. In Line 216, what is R2 for strong correlation between FDG uptake and macrophage/lymphocytic cellular infiltration?

This has been amended

12. On Line 220, what do you mean by "Metabolic activity". Please clarify.

This has been amended to state glucose metabolism

13. In Line 223, why R2 = 0,19 is presented as a modest correlation? What is the range for poor, modest and strong correlation in terms of R2 values?

This has been amended

14. In Line 230, do you mean "tracer-specific activity" instead of "tracer-specific factors"?

Thank you. I have reviewed this sentence and it is written as intended.

15 Please add equation of both SUV and TBR to mathematically show how TBR calculation avoids consideration of patient's weight, injected dose and imaging time point.

This has been added.

16. What does author think if SUV or TBR should be used?

Thank you. We have elaborated on this in the revised manuscript and commented on the preferred use of SUV over TBR for the purposes of risk stratification after stroke.

17. In entire review please clarify in publications cited the high PET signal or high uptake when comparing was quantified as SUV or TBR? For example, in Line 330-332, is FDG uptake quantified as SUV or TBR?

This has been amended

18. Provide citation to Line 339-341 and line 358-362.

This has been amended

19. In Line 405, provide citation for Gionnotti et al.

This has been amended

20. In SCAIL Points in Table 1 why TBR is not used if TBR is preferred over SUV?

Thank you. SUV was used as it had more robust and stronger associations with recurrent stroke than FDG.

21. Please provide TBR or SUV values when mentioning uptake of different tracers in section 10 for novel tracers, For example nu uptake values are given for DOTATATE in Line 559-567.

This section has been edited accordingly

22. Please discuss studies where FDG is directly compared with the other novel tracers discussed in section 10.

This section has been edited accordingly. These edits apply to the discussion about NaF. Limited direct comparisons exist vs FDG exist for other tracers.

Reviewer 2 Report

I read a comprehensive review regarding PET imaging in carotid atherosclerosis and stroke, I have only minor comments:

1. In the technical issues section I would report more regarding the problems and potential solutions in FDG PET imaging in diabetes mellitus. 

2. A graph depicting the atherosclerosis process (macrophage infiltration, intraplaque hemorrhage, microcalcification) and potentially the applications of PET traces would be very helpful

Author Response

Thank you for your comments. Please see below the reply.

  1. In the technical issues section I would report more regarding the problems and potential solutions in FDG PET imaging in diabetes mellitus. 

I have addressed this in the revised version.

  1. A graph depicting the atherosclerosis process (macrophage infiltration, intraplaque hemorrhage, microcalcification) and potentially the applications of PET traces would be very helpful

I have created a new figure (figure 1)

Reviewer 3 Report

The authors provide a well elaborated review about tracers for imaging carotid plaque inflammation.

The authors provide a detailed, well structured and extremely elaborated review about PET imaging options in nuclear medicine for carotid plaque inflammation imaging. They give an overview about the standard FDG-imaging, its benefit and limitations and provide information about other tracers benefitial for this purpose.
Up to now there is no specific guideline whcih address this purpose, although one is at the moment in preparation by the respective commitee of the EANM. The review adress this gap, collecting all already published information and setting it into context to the drafted guideline.
It provides a short overview which collects all relevant publications, which makes it easier for the reader to figure out relevant data and advices.

The conclusions are consistent with the evidence and arguments presented and they do address the main question posed? The reference list is very elaborate and covers from my point of view the relevant publications in that specific field. The format of the list should be adjusted, but this is a minor flaw, which I think is in any case corrected before publication. The figures are of sufficient quality and substantiate the thesis.

Before publication the manuscript should be thoroughly revised with regard to nomenclature of radiopharmaceuticals and radionuclides. 

Author Response

Thank you for your comments. I have updated the manuscript formatting accordingly.